# Cytomegalovirus Pneumonia in a Patient with X-Linked Agammaglobulinemia: A Case Report

**DOI:** 10.3390/medicina58101457

**Published:** 2022-10-15

**Authors:** Yao-Xian Wong, Shyh-Dar Shyur

**Affiliations:** 1Department of Pediatrics, Mackay Memorial Hospital, Taipei 104, Taiwan; 2Department of Pediatrics, Far Eastern Memorial Hospital, New Taipei City 220, Taiwan; 3Mackay Medical College, New Taipei City 25245, Taiwan

**Keywords:** case report, cytomegalovirus pneumonia, X-linked agammaglobulinemia

## Abstract

X-linked agammaglobulinemia (XLA) is a hereditary immune disorder that predisposes patients to frequent and severe bacterial infections caused by encapsulated bacteria (such as *Streptococcus pneumoniae*, *Staphylococcus aureus*, and *Haemophilus influenzae*). Otitis media, sinusitis, and pneumonia are common complications of XLA that require prompt diagnosis and treatment. Cytomegaloviruses (CMV) cause widespread and severe infections in immunocompromised individuals, affecting the respiratory tract, and consequently, leading to pneumonia, which is associated with a high mortality rate. However, CMV-induced pneumonia is rarely reported in patients with XLA. This case study details a 37-year-old male patient with XLA presenting with fever, productive cough, and dyspnea. The patient was diagnosed with CMV pneumonia and recovered after treatment. To the best of our knowledge, this is the first reported case of CMV pneumonia in a patient with XLA in Taiwan. This case study emphasizes that CMV pneumonia in patients with XLA is a treatable condition if diagnosed promptly, and that a shorter duration of treatment with the antiviral agent, in combination with immunoglobulin replacement therapy, can resolve symptoms.

## 1. Introduction

X-linked agammaglobulinemia (XLA) is one of the most frequent inborn errors of immunity, with an estimated prevalence of 3–6 per 1,000,000 people in Taiwan. XLA is caused by mutations in the Bruton tyrosine kinase (*BTK*) gene on the X chromosome (Xq21.3–Xq22), resulting in profound B-cell lymphopenia and agammaglobulinemia in most cases. XLA is an X-linked recessive disease that primarily affects males, with females being asymptomatic carriers. Patients with XLA in infancy or childhood present with repeated microbial infections typically caused by encapsulated bacteria and a complete absence of tonsils and lymph nodes. The most common pathogens responsible for these infections are *Haemophilus influenzae*, *Streptococcus pneumoniae*, *Pseudomonas aeruginosa*, and *Moraxella catarrhalis* [1]. However, there are patients with atypical XLA characterized by less invasive infections, who have responded well to antibiotics treatment, showing near-normal levels of serum immunoglobulin, and in some cases, presented with small cervical lymph nodes [2,3]. Patients with atypical XLA are initially misdiagnosed as harboring other humoral deficiency diseases, such as transient hypogammaglobulinemia of infancy [4], specific antibody deficiency [5], common variable immunodeficiency [6], or selective immunogloblulin M deficiency [3]. Despite no curative treatment being available currently for XLA, patients are administered immunoglobulin replacement therapy (IRT) and prophylactic antibiotics to prevent recurrent infections. Notably, IRT is beneficial in reducing the risk of pulmonary infections, subsequently increasing the life expectancy of patients with XLA.

Cytomegalovirus (CMV) is a double-stranded DNA virus and a member of the *Herpesviridae* family. CMV is widely considered as an infectious pathogen responsible for life-threatening community-acquired pneumonia in immunocompromised individuals, being characterized by respiratory distress, dyspnea, and cough. CMV is diagnosed using virological, histopathological, or immunochemical analyses of biopsied lung tissue. CMV is commonly detected by viral isolation, rapid culture of bronchoalveolar lavage (BAL) fluid, and quantitation of CMV DNA in BAL fluid [7]. The present study describes a case of CMV pneumonia in a Taiwanese patient with XLA. This case study clarifies the importance of early diagnosis of infections caused by opportunistic pathogens, such as CMV, in patients with primary hypogammaglobulinemia, to ensure prompt and efficient treatment.

## 2. Case Report

A 37-year-old male with XLA visited our hospital with a three-day history of fever, productive cough, and dyspnea. The patient had been administered intravenous immunoglobulins (IVIG) every 4 weeks and prophylactic antibiotics with trimethoprim/sulfamethoxazole since XLA diagnosis at age 11. The results of lymphocyte subset analysis revealed CD19^+^ 1.8% (range 8–32%) in the peripheral blood, and the serum immunoglobulin (Ig) levels were as follows: IgG, 7 mg/dL (range, 1097–1518 mg/dL); IgM, 10 mg/dL (range, 95–188 mg/dL); and IgA, 3 mg/dL (range, 142–277 mg/dL). Direct sequencing mutation analysis revealed a *BTK* gene defect c.839+4 A>G (p.E280 fsx 281) mutation. The patient had recurrent sinusitis, acute otitis media with eardrum perforation, *Pseudomonas* pneumonia with empyema, and meningoencephalitis; there was no history of a previous CMV infection. Upon initial examination, the patient was febrile (38.3 °C) with a heart rate, respiratory rate, and blood pressure of 122 beats/min, 30 breaths/min, and 99/62 mmHg, respectively. Other physical examinations were within normality; however, lung auscultation of the patient revealed diffuse wheezing breath sounds with fine crackles in the lower lung zones. Blood test results revealed significant leukocytosis (21,700 cells/µL), with a neutrophil proportion of 81%, and elevated C-reactive protein levels (14 mg/dL; range, <0.5 mg/dL). The patient had received IVIG three weeks prior to the tests. Serum immunoglobulin (Ig) levels were as follows: IgG, 585 mg/dL (range, 751–1560 mg/dL); IgM, <25 mg/dL (range, 46–304 mg/dL); and IgA, <6 mg/dL (range, 82–453 mg/dL). Real-time polymerase chain reaction (RT-PCR) results for SARS-CoV-2 infection were negative. Chest X-ray evaluation revealed interstitial linear infiltrates in bilateral lower lung fields and right upper lung fields (Figure 1A). The arterial blood gas had a pH of 7.34, pCO_2_ of 45 mmHg, and pO_2_ of 68 mmHg. Upon admission, chest computed tomography imaging revealed multifocal peribronchial infiltrates, centrilobular nodules, and patchy consolidation in the upper lung field of both lungs (Figure 1C,D). No organisms were detected in blood cultures, sputum cultures, or sputum smear microscopy. Urine samples were negative for *Legionella* and *Streptococcus pneumoniae* antigens. Although the patient was administered a 3-day course of empiric antibiotics with intravenous meropenem, linezolid, and ganciclovir 5 mg/kg twice daily and a 2-day course of voriconazole, his clinical condition continued to deteriorate. Oral tracheal intubation was performed, and invasive mechanical ventilation was provided on day 3. A chest X-ray was performed after intubation (Figure 1B).

The patient received IVIG treatment on day 3, which subsequently led to an increase in serum IgG concentration. Serum IgG increased to 1173 mg/dL (range, 751–1560 mg/dL) after IVIG. Flexible bronchoscopy followed by BAL was performed on day 3 after intubation; however, no endobronchial lesion was observed. The BAL specimen did not show any malignant cells; however, cytological analysis revealed neutrophilia. BAL fluid cultures and PCR tests were negative for toxic components secreted by fungal organisms (e.g., galactomannan) and for infections by *Pneumocystis jirovecii*, *Mycoplasma pneumoniae*, herpes simplex virus, and *Mycobacterium tuberculosis*. Despite administering five doses of ganciclovir treatment, CMV was isolated from a BAL fluid viral culture, and PCR analysis revealed a CMV viral load of 497,500 copies/mL (range < 250 copies/mL) in the BAL fluid. Subsequently, PCR analysis of both plasma and sputum samples revealed a CMV viral load of 3 log IU/mL and 368,800 copies/mL (range < 625 copies/sample), respectively. The mechanical ventilation was successfully retracted after 5 days. Intravenous ganciclovir transitioned to oral valganciclovir 900 mg daily on day 6. The patient received a 7-day course of the antibiotics therapy (meropenem, linezolid) and 2-day course of voriconazole. The patient, who was admitted for 12 days, was discharged after treatment.

## 3. Discussion

Herein, we described the case of a patient with XLA presenting symptoms of CMV pneumonia, which was confirmed by serology and RT-PCR analysis of BAL fluid and sputum. To date, only two studies have reported CMV as the pneumonia-causing pathogen in patients with XLA (Table 1) [8,9]. In these case studies, the patients were treated with antibiotics and IVIG, demonstrating treatable outcomes with and without ganciclovir therapy. Ganciclovir and valganciclovir are the most commonly used drugs for treating CMV infection [10]. Systemic treatment or prevention of CMV infection in immunocompromised patients is induction with a 7 to 14-day course of ganciclovir 5 mg/kg twice a day. For maintenance after transplant, a 100 to 120-day course of 5 mg/kg or induction with a 21-day course of valganciclovir 900 mg twice a day and maintenance with 900 mg once per day are recommended [11,12,13]. In immunocompetent individuals with CMV pneumonia, valganciclovir 900 mg was administered twice a day for at least 14 days for symptom resolution [14]. A combination of ganciclovir and IVIG is the recommended therapy for CMV pneumonia in transplant recipients [15]. IRT may not significantly reduce viral respiratory infections among patients with primary antibody deficiency [16]; however, CMV was not included in the study. IVIG treatment reduces intracellular virus production and viral protein synthesis, eventually limiting the spread of CMV; thus, IVIG may contribute to the control and alleviation of CMV infection [17]. Moreover, IVIG showed similar CMV neutralization effects as CMV-specific hyperimmunoglobulins [18]. Significant clinical improvement after IVIG was observed, and mechanical ventilation support was retracted on day 5 of admission for the patient.

CMV is commonly found in BAL fluid samples due to asymptomatic shedding; therefore, higher levels of CMV DNA or CMV viral load in BAL fluids than in plasma are suggestive of CMV pneumonia [19]. Moreover, a study demonstrated that a cut-off value of CMV DNA levels >500 IU/mL in BAL fluid can be used to differentiate between CMV pneumonia and pulmonary shedding with a positive predictive value of 45% [20].

In the present case, despite the elevated levels of serum IgG, other common pathogenic microorganisms were absent compared with two other studies [8,9], and only CMV was detected. Serum IgG trough levels >500 mg/dL are generally accepted as the required minimum threshold for patients with primary immunodeficiency [21]. Because IgG trough levels differ between patients, IRT may be personalized based on individual factors, such as clinical symptoms and infections, to maximize the outcome benefits [22,23]. IRT can be administered intravenously (IVIG) or subcutaneously (SCIG), and both have similar efficacy and safety profiles [24,25]; the administration of IVIG results in a rapid peak in IgG concentration that gradually declines before the next infusion, whereas SCIG provides more consistent IgG levels. Weekly subcutaneous administration of SCIG was shown to result in higher serum IgG trough levels and lower infection rates than with IVIG administration [25]. Therefore, SCIG may be an alternative IRT strategy for the present case.

*BTK* mutations are responsible for the increased susceptibility to respiratory infections in patients with XLA. *BTK* is involved in the Toll-like receptor (TLR)-8/9 signaling pathways, which are important for the activation of host defenses against bacterial and viral infections [26]. Hence, *BTK* mutations in patients with XLA lead to impaired activation of TLRs, and consequently, to decreased production and secretion of proinflammatory cytokines, such as tumor necrosis factor-alpha and interleukin (IL)-6 [26,27]; this, in turn, may contribute to increased susceptibility to respiratory viral infections in these patients. The recurrence of CMV pneumonia in patients with XLA needs long term follow-up.

## 4. Conclusions

This case study aims to emphasize that CMV pneumonia in patients with XLA is a treatable condition. It also clarifies the importance of early diagnosis of opportunistic pathogen infections, such as CMV, occurring despite regular IRT, to ensure prompt and efficient treatment. As IRT is effective against CMV, a shorter duration of antiviral treatment (ganciclovir or valganciclovir) or IRT as monotherapy may suffice for symptom resolution in these patients. Further research evaluating the efficacy of IVIG for routine clinical use in the treatment of CMV pneumonia in patients with XLA is imperative.

## Figures and Tables

**Figure 1 medicina-58-01457-f001:**
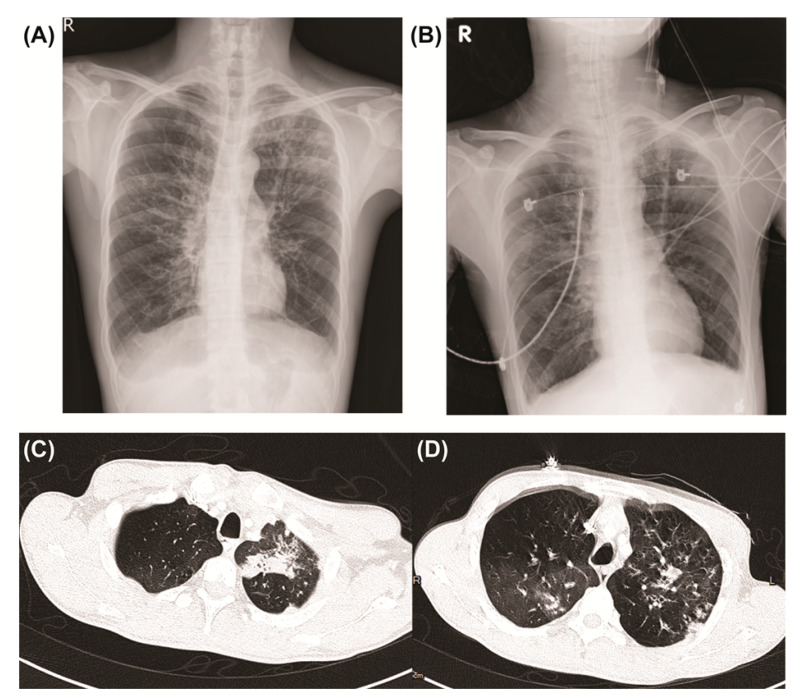
Chest X-ray and computed tomography images. (**A**,**C**,**D**) On the day of admission to our hospital, and (**B**) after intubation.

**Table 1 medicina-58-01457-t001:** Available studies on cytomegalovirus pneumonia in patients with X-linked agammaglobulinemia.

Ref.	Patient Age (Years)	Serum IgG (Normal Range)	Pathogen	Treatment	Outcome
Arroyo-Martinez et al. [8]	20	<108 (540–1822 mg/dL)	CMV, RSV, and rhinovirus	Vancomycin, piperacillin/tazobactam, and IVIG	Discharged
Xu et al. [9]	22	5.55 g/L (7.23–16.85 g/L)	*Staphylococcus epidermidis*, *Candida albicans*, and CMV	Teicoplanin, voriconazole, ganciclovir, and IVIG	Discharged

Abbreviations: CMV, cytomegalovirus; IgG, immunoglobulin G; IVIG, intravenous immunoglobulins; RSV, respiratory syncytial virus.

## Data Availability

Not applicable.

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
