# Peer review of "Cytomegalovirus Pneumonia in a Patient with X-Linked Agammaglobulinemia: A Case Report"

_medicina, 2022, doi:10.3390/medicina58101457_

Round 1

Reviewer 1 Report

Dear authors, I read your paper and have the following comments:

MAJOR COMMENTS:

-this paper is a case report, as stated in the title. It is therefore not a study per se. If you want to keep the words study you can call it case study. Please amend accordingly

-in the abstract you state that XLA predisposes to frequent and severe bacterial infections. I do agree, but it may be good to qualify the statement explaining that infections with encapsulated bacteria are more likely.

-I do not agree that based on this case report it can be derived that CMV pneumonia in patients with XLA has a favourable outcome. The level of evidence is simply too low. Please revise.

-In the introduction I suggest that you explain that XLA can be typical, but also atypical. There are differences in the Ig levels and clinical manifestations between the two. Prophylactic antibiotics can be considered, yes, but I do not think this is a must.

-in the case report, nothing is mentioned about the XLA diagnosis. Was this typical or atypical? When was the diagnosis made? What previous infections did the patient have? What treatment did he receive and since when? I am not a radiologist, but are we sure that there were no differences at all between chest Xray before and after intubation? When was the patient intubated, how soon after admission and after how many doses of therapy? Why was IVIG only given at day 3? Was the BAL CMV viral load measured on the first day of admission or later? After how many doses of ganciclovir? What was the total duration of treatment with each substance and how did the clinical signs and symptoms evolve? All these need to be presented

-conclusions: as per my comment before, you cannot state based on this evidence that CMV pneumonia in XLA patients has a favourable outcome. You need to clarify what you mean by a shorter ganciclovir therapy. What dose and duration and based on what evidence?   

Reviewer 2 Report

Abstract: the content is clear and concise. An introduction to XLA and CMV is logically provided and the key features of the clinical case described are then briefly explained.

Introduction: As expected, it addresses the characteristics of XLA and CMV. It also explains the content and purpose of the article clearly.

Case Report: the clinical case is described with precise references to blood chemistry values ​​and initiated therapies. I appreciated the presence of radiological images, which allow a more precise evaluation of the picture. The exposure is always clear and fluid.

Discussion:  treatment of CMV infection is clarified and the rarity of this in the course of XLA is emphasized.

Conclusion: the positive outcome of this infection is underlined, explaining how an early diagnosis is important. I appreciated the indication for further future research to evaluate the effectiveness of intravenous therapy as a clinical routine in CMV infection.

Author Response

Thank you for your comment.

Round 2

Reviewer 1 Report

Thanks for the revision; the paper has gained in clarity